# Quality of Life in Rectal Cancer Treatments: An Updated Systematic Review of Randomized Controlled Trials (2013–2023)

**DOI:** 10.3390/cancers17142310

**Published:** 2025-07-11

**Authors:** Silvia Negro, Francesca Bergamo, Lorenzo Dell’Atti, Alessandra Anna Prete, Sara Galuppo, Marco Scarpa, Quoc Riccardo Bao, Stefania Ferrari, Sara Lonardi, Gaya Spolverato, Emanuele Damiano Luca Urso

**Affiliations:** 1Third Surgery Unit, Department of Surgical, Oncological, and Gastroenterological Sciences, University of Padua, 35128 Padua, Italy; l.dellatti@gmail.com (L.D.); marco.scarpa@unipd.it (M.S.); stefania.ferrari10@gmail.com (S.F.); gaya.spolverato@unipd.it (G.S.); edl.urso@unipd.it (E.D.L.U.); 2Medical Oncology Unit 1, Veneto Institute of Oncology (IOV-IRCCS), 35128 Padua, Italy; francesca.bergamo@iov.veneto.it (F.B.); alessandraanna.prete@iov.veneto.it (A.A.P.); sara.lonardi@iov.veneto.it (S.L.); 3Radiotherapy Unit, Veneto Institute of Oncology (IOV-IRCCS), 35128 Padua, Italy; sara.galuppo@iov.veneto.it

**Keywords:** health-related quality of life, rectal cancer, organ-preserving strategies, radiotherapy

## Abstract

As survival outcomes for rectal cancer continue to improve, the long-term impact of treatment on quality of life has become an essential consideration in clinical decision-making. This systematic review synthesizes evidence from 41 randomized controlled trials conducted between 2013 and 2023, encompassing over 9000 patients. Organ-preserving surgical strategies—particularly sphincter-sparing and rectum-sparing procedures—were consistently associated with superior quality of life compared with more radical approaches such as abdominoperineal resection. While radiotherapy remains a cornerstone of oncological control, it was frequently linked to adverse effects on bowel, sexual, and general functional outcomes. Patient-centered interventions, including structured continuity-of-care models and techniques such as transanal irrigation, demonstrated potential in mitigating postoperative dysfunction, especially in patients with ostomies or low anterior resection syndrome. These findings highlight the need for a more nuanced and individualized approach to rectal cancer management that balances oncological efficacy with long-term functional preservation.

## 1. Introduction

Colorectal cancer (CRC) is the third most common malignancy worldwide, with rectal cancer accounting for approximately one third of all CRC cases. In recent decades, mortality rates for both CRC and rectal cancer have decreased significantly in Europe and the United States [1,2], mainly because of advances in therapeutic strategies and improvements in early detection. As survival rates improve, integrating health-related quality of life (HRQoL) considerations into clinical management has become increasingly important. The treatment of rectal cancer requires a multidisciplinary approach that combines complex surgical planning, preoperative and postoperative chemotherapy (CT) and/or radiotherapy (RT), and comprehensive patient care strategies. All of these treatments can have a significant impact on functional outcomes and patients’ HRQoL [3,4,5,6]. For rectal cancer, the standard surgical treatment typically consists of total mesorectal excision (TME), with the extent of rectal surgery being a critical factor influencing HRQoL [7]. Patients who undergo ostomy creation and abdominoperineal resection (APR) typically report a decreased HRQoL, although some exceptions exist [8]. Preoperative CT and/or RT are recommended to improve local disease control in patients with locally advanced middle and low rectal cancers [9]. However, several randomized clinical trials (RCTs) and a 2018 Cochrane review have shown that neoadjuvant RT is associated with a higher risk of sexual dysfunction and fecal incontinence compared to surgery alone [10,11,12,13]. In light of these findings, researchers have focused on evaluating various therapeutic adjuncts, including psychosocial support, nutritional interventions, and pelvic-floor rehabilitation programs, to improve functional outcomes. Despite a growing body of the literature examining the individual impact of these treatments on HRQoL, a definitive international consensus on the complex interplay between different treatment modalities is still lacking. Moreover, the use of various questionnaires to assess HRQoL complicates the comparison of results across trials [14]. Therefore, the present study aimed to address this knowledge gap by conducting a systematic review and synthesis of high-quality, evidence-based studies published in the last 11 years. The focus will be on RCTs that assess the impact of surgery, preoperative and postoperative CT/RT, and patient care strategies on HRQoL in rectal cancer patients.

## 2. Materials and Methods

### 2.1. Study Registration

The study protocol was registered in the International Prospective Register of Systematic Reviews (PROSPERO) on 15 January 2024. Ethical review and approval were waived for this study due to the article being a systematic review of randomized controlled trials.

### 2.2. Search Strategy

A literature search was conducted to identify studies reporting HRQoL in patients with rectal cancer. A systematic review of Scopus, EMBASE, MEDLINE, and the Cochrane Central Register of Controlled Trials was conducted using the following Medical Subject Headings (MeSH) search terms: “rectal neoplasms” AND “quality of life” OR “functional outcomes.” Articles published between 1 January 2013 and 31 December 2023 were included, along with the references of relevant articles (Appendix A).

We selected the 2013–2023 time frame to focus on RCTs that reflect contemporary treatment practices, HRQoL measurement tools, and supportive care strategies. Treatments, surgical techniques, and patient-centered outcomes have evolved significantly in the past decade, and including older studies might have introduced outdated or less relevant practices. Two independent reviewers (S.N. and L.D.) conducted the literature review separately according to the established inclusion criteria. Data regarding the year of publication, country of study, duration of the study, number of patients enrolled, intervention, QoL questionnaire used, duration of follow-up, and QoL outcomes were extracted and registered separately by the reviewers, and a database of the selected papers was created. After duplicates were removed, disagreements were resolved by two additional blinded reviewers (F.B. and E.D.L.U.). The systematic review was conducted in accordance with the Preferred Reporting Items for Systematic Reviews and Meta-Analyses (PRISMA) guidelines [14]. Given the nature of the study (systematic review) and the absence of direct reporting of individual patient data, Institutional Review Board approval was not required. The work conducted has been reported in line with AMSTAR (Assessing the methodological quality of systematic reviews) guidelines (Appendix A) [15].

### 2.3. Inclusion and Exclusion Criteria

All randomized controlled trials (RCTs) that reported HRQoL in patients with rectal cancer between January 2013 and December 2023 were considered eligible for inclusion. HRQoL data can be collected through self-administered questionnaires or structured interviews (by telephone or in person). Controlled clinical trials, and those that were observational, retrospective, prospective, cohort, population-based, cross-sectional, case-control, editorials, reviews, invited commentaries, and case reports were excluded. Only RCTs were included to ensure high methodological quality, minimize bias, and provide robust evidence for causal relationships between interventions and outcomes. Only studies published in English that specifically focused on rectal cancer or clearly distinguished rectal and colon cancer were included.

### 2.4. Data Collection and Analysis

Studies on HRQoL and rectal cancer were categorized into three key areas of interest: surgical intervention, pre-and/or post-CT and/or RT, and patient care strategies. The term “Radiotherapy” was used as a MeSH term in our research to specifically capture therapeutic interventions, while the broader term “radiation” was excluded to maintain focus and specificity. Patient care strategies are defined as the range of interventions and practices designed to address the physical, psychological, emotional, and social needs of patients. These include continuity-of-care programs, education and counseling, symptom management, rehabilitation services, psychosocial support, and lifestyle modifications. Separate tables were created for each category, and data were presented and analyzed using pro forma tables with prespecified variables. The dataset included the following variables: study design, period, population, intervention, number of patients, age, sex, distance from the anal verge, outcome measures used, duration of follow-up, and outcomes. A meta-analysis was not performed because of significant heterogeneity among the included studies in terms of design, population characteristics, interventions, HRQoL measurement tools, and reported outcomes. Therefore, funnel plots could not be used to assess publication bias in this context. Because of this variability, calculating a quantitative heterogeneity index (e.g., I^2^) was inappropriate because it would not reliably measure inconsistency across studies with such wide-ranging methodological approaches. Thus, a structured narrative synthesis was adopted to clarify the findings and maintain their clinical relevance.

### 2.5. Quality Assessment

The quality of eligible RCTs was assessed using the Cochrane Risk-of-Bias Tool (RoB 2). This involved the evaluation of the following domains: randomization sequence generation, allocation concealment, blinding of participants and personnel, completeness of outcome data, selective outcome reporting, and other potential sources of bias. Each domain was assessed to rate each RCT as having a low, high, or unclear risk of bias.

## 3. Results

### 3.1. Study Selection

A total of 1004 articles was initially identified, 309 of which were selected for full-text review. After excluding unavailable and irrelevant articles, 41 studies were selected for the systematic review (Figure 1). Inter-rater agreement between reviewers was strong, with a Cohen’s kappa (κ) of 0.79. Among the included studies, 16 focused on surgical interventions (3507 patients), 15 on pre-and/or post-CT and/or RT (5114 patients), and 10 on patient care strategies (619 patients).

### 3.2. Synthesis of Clinicopathological Data

Table 1, Table 2 and Table 3 (and Appendix A) summarize 41 studies that met the eligibility criteria. Of these, 21 were multicenter RCTs, and 10 were single-center RCTs. A variety of HRQoL measurement tools were used across the studies: cancer-specific tools (the European Organization for Research and Treatment for Cancer (EORTC)) QLQ30), colorectal cancer-specific tools (EORTC CR-38, EORTC QLQ-CR29, the Functional Assessment of Cancer Therapy-Colorectal-C (FACT-C)), a stoma-specific tool (Stoma-QoL, designed for patients living with a stoma), generic quality-of-life questionnaires (36-Item Short Form (SF-36) and the EuroQoL EQ-5D), and a physical and mental health-specific tool (12-Item Short Form (SF-12). The reviewed studies reported patient-reported outcome (PRO) data collected at various follow-up stages, ranging from the early postoperative period (e.g., four weeks) to long-term assessments (e.g., 12 years).

#### 3.2.1. Surgical-Intervention Studies

A total of 3507 patients with rectal cancer were included in 16 studies on surgical interventions [16,17,18,19,20,21,22,23,24,25,26,27,28,29,30,31] (Table 1). Of these patients, 47% to 74% were male, and their ages ranged from 30 to 90 years. The follow-up period ranged from 4 to 12 years. Three studies used the EORTC QLQ-CR38, seven used the EORTC QLQ-CR30, two used the FACT-C, two used the EORTC QLQ-C30, one used the SF-36, one used the SF-12, and one used the Stoma-QOL.

#### 3.2.2. Pre- and/or Post-CT and/or RT Studies

A total of 5114 patients with rectal cancer were included in 15 studies on pre- and/or post-CT and/or RT [32,33,34,35,36,37,38,39,40,41,42,43,44,45,46] (Table 2). Of these patients, 47% to 72% were male, and their ages ranged from 22 to 84 years. The follow-up period ranged from 12 years to 14 years. Twelve studies used the EORTC QLQ-C30, five used the EORTC QLQ-CR38, five used the EORTC QLQ-CR29, one used the SF-36, one used the FACT-C, and one used the EuroQoL EQ-5D.

#### 3.2.3. Patient Care Strategies Studies

A total of 619 patients with rectal cancer were included in 10 studies on patient care strategies [47,48,49,50,51,52,53,54,55,56] (Table 3). Of these, 5 to 64% were male, and their median age ranged from 57 to 71 years. The follow-up period ranged from 3 weeks to 2 years. Four studies used the EORTC QLQ-C30, five used the EORTC QLQ-CR29, one used the SF-36, one used the SF-12, and one used the Stoma-QoL.

**Table 1 cancers-17-02310-t001:** Articles on surgical interventions included in the review.

Author, Year [Ref.]	Duration of Study	Population	Intervention	Number of Patients	Questionnaire(s)	Times of Assessment	Key Results
Andersson et al., 2013 [16]	2004–2010	To compare HRQoL after laparoscopic vs. open surgery in patients undergoing LAR	Laparoscopic (*n* = 260) vs. Open Surgery (*n* = 125)	385	EORTC-QLQ CR38 EORTC-QLQ-C30	4 w, 6, 12 and 24 mo after operation	No difference
Russell et al., 2015 [17]	Jul 2004–Aug 2010	To compare HRQoL after SS surgery vs. APR	SS surgery (*n* = 615) vs. APR (*n* = 372) *SS: Sphincter Saving Surgery* *APR: Abdominoperineal Resection*	987	FACT-C, EORTC QLQ CR38	12 mo after operation	Worse body image, sexual activity (only for males), micturition symptoms and GI tract symptoms for the APR group
Okkabaz et al., 2017 [18]	Jun 2009–N/A	To compare HRQoL after colonic j-pouch vs. side-to-end anastomosis	CJP (*n* = 29) vs. SCA (*n* = 28) *CJP: Colon J-Pouch anastomosis* *SCA: Straight Colorectal anastomosis*	57	SF-36	4, 8, 12 mo after stoma reversal	No difference
Musters et al., 2017 [19]	Feb 2013–Sep 2014	To compare HRQoL after mesh closure vs. primary closure after APR	Mesh closure (*n* = 48) vs. Primary closure (*n* = 53)	101	EORTC-QLQ CR29 EORTC-QLQ-C30 SF36	3, 6, 9, 12 mo after the operation	No difference
Gadan et al., 2017 [20]	Dec 1999–June 2005	To compare HRQoL after temporary ileostomy vs. no ileostomy in patients undergoing LAR	Temporary ileostomy (*n* = 41) vs. no ileostomy (*n* = 46)	87	EQ-5D-3L	12 y after the operation	Worse self-reported HRQoL in those with major LARS
Jayne et al., 2017 [21]	Jan 2011–Sept 2014	To compare HRQoL after robotic vs. laparoscopic surgery in patients undergoing LAR	Robotic Surgery (*n* = 237) vs. Laparoscopic surgery (*n* = 234)	471	SF 36	30 days, 6 mo after the operation	No difference
Kim et al., 2018 [22]	Feb 2012–March 2015	To compare HRQoL after robotic vs. laparoscopic surgery in patients undergoing LAR	Robotic Surgery (*n* = 66) vs. Laparoscopic surgery (*n* = 73)	139	SF 36	3 we, 3 mo, 12 mo after the operation	No difference
Park et al., 2018 [23]	Feb 2011–Nov 2015	To compare HRQoL after early vs. late ileostomy closure after LAR	Early ileostomy closure (8–13 days after surgery) (*n* = 55) vs. late ileostomy closure (>12 we after surgery) (*n* = 57)	112	EORTC-QLQ CR29 EORTC-QLQ-C30 SF36	3, 6, 12 mo after the operation	No difference
Parc et al., 2019 [24]	2007–2009	To compare HRQoL after colonic j-pouch vs. side-to-end anastomosis	CJP (*n* = 80) vs. SEA (*n* = 87) *SEA: Side-to-End anastomosis* *CJP: Colon J-Pouch anastomosis*	167	SF12, FACT-C	6, 12 and 24 mo after surgery	No difference
Ribi et al., 2019 [25]	Sep 2005–May 2014	To compare HRQoL after colonic j-pouch vs. side-to-end vs. straight colorectal anastomosis	CJP (*n* = 63) vs. SEA (*n* = 95) vs. SCA (*n* = 99) *SEA: Side-to-End anastomosis* *CJP: Colon J-Pouch anastomosis* *SCA: Straight Colorectal anastomosis*	257	FACT-C	6, 12, 28 and 24 mo after surgery	No difference
Gavaruzzi et al., 2020 [26]	Oct 2009–Feb 2016	To compare HRQoL after colonic j-pouch vs. straight colorectal anastomosis	CJP (*n* = 161) vs. SCA (*n* = 158) *CJP: Colon J-Pouch anastomosis* *SCA: Straight Colorectal anastomosis*	319	EORTC QLC C30 EORTC QLC C38	6, 12, 24 mo after the operation	No difference
Bach et al., 2020 [27]	Feb 2012–Dec 2014	To compare HRQoL after organ preservation (LE) vs. radical surgery in cT2, or lower RC, N0, M0, who underwent short course RT	Organ preservation (LE) (*n* = 27) vs. radical surgery (*n* = 28)	56	EORTC QLC C30 EORTC QLC CR29	3, 6, 12, 24 and 36 mo after the operation	Worse QoL in the areas of health anxiety, role, and social function for the organ-preservation group
Elsner et al., 2021 [28]	Nov 2007–Mar 2014	To compare HRQoL after early vs. late ileostomy closure after LAR	Early ileostomy closure (2 we) (*n* = 37) vs. late ileostomy closure (12 we) (*n* = 34)	71	EORTC- QLQ-C30	6 we and 4 mo after surgery	No difference
Dulskas et al., 2021 [29]	Dec 2011–Dec 2017	To compare HRQoL after early vs. late ileostomy closure after LAR	Early ileostomy closure (30 days) (*n* = 26) vs. late ileostomy closure (90 days) (*n* = 25)	51	EORTC- QLQ-C30	36 mo	No difference
Ellebaek et al., 2023 [30]	Apr 2011–Sep 2018	To compare HRQoL after early vs. late ileostomy closure after LAR	Early ileostomy closure (8-12 days) (*n* = 77) vs. late ileostomy closure (>3 mo) (*n* = 69)	146	GIQLI	6 mo, 12 mo after the operation	No difference
Ahmadi-Amoli et al., 2023 [31]	2016–2020	To compare HRQoL after early vs. late ileostomy closure after LAR	Early ileostomy closure (2–3 we after the first two courses of adjuvant chemotherapy) (*n* = 50) vs. late ileostomy closure (2–3 we after the last course of adjuvant chemotherapy) (*n* = 54)	104	SF 36	3, 12 mo after the operation	No difference

LAR: low anterior resection; HRQoL: health-related quality of life; LARS: low anterior resection syndrome.

**Table 2 cancers-17-02310-t002:** Articles on pre- and/or post-CT and/or RT included in the review.

Author, Year [Ref.]	Duration of Study	Population	Intervention	Number of Patients	Questionnaire(s)	Times of Assessment	Key Results
McLachlan et al., 2016 [32]	2001–2006	To compare HRQoL after short-course RT vs. long-course CRT in patients with cT3N0-2M0	Short-course RT (*n* = 143) vs. Long-course CRT (*n* = 154)	297	EORTC QLQ C30—QLQC38	1, 2, 3, 6, 9, 12 mo after treatment	No difference
Wiltink et al., 2016 [33]	Jan 1996–Dec 1999	To compare HRQoL after short course RT followed by TME vs. TME alone in patients with LARC planned for surgery	Short-course RT (study group, 5 × 5 Gy) followed by TME (*n* = 241) vs. TME alone (control group, *n* = 237)	478	EORTC QLQ C30—QLQCR29	3, 6, 12, 18, 24 mo, 5 y and 14 y	More bowel disfunction in the RT group
Araujo et al., 2018 [34]	Jan 2011—Feb 2013	To compare HRQoL after nCRT (capecitabine) vs. nCRT (5-FU and leucovorin) in patients with stage II and III RC planned for surgery	nCRT (Group 1, capecitabine, *n* = 31) vs. nCRT (Group 2, 5-FU and leucovorin, *n* = 30)	61	EORTC QLQ C30—QLQCR38	6–8 we after nCRT, 30 days after surgery, after adjuvant CT, 1 year after the end of the treatment or stoma closure	No difference
Sang Hong et al., 2019 [35]	Nov 2008–Jun 2012	To compare HRQoL after adjuvant CT with Fl vs. adjuvant CT with FOLFOX in patients with ypStage II or III RC	Adjuvant CT with FL (fluorouracil and leucovorin) (study group, *n* = 161) vs. adjuvant CT with FOLFOX (control group, *n* = 160)	321	EORTC QLQ C30—QLQCR38	2 mo, and at the end of treatment	No difference
Van der Valk et al., 2019 [36]	2004–2013	To compare HRQoL after adjuvant CT in patients with ypStage II or III RC who underwent preoperative (CT)RT	Adjuvant CT (Capecitabine, 8 courses) (study group, *n* = 115) vs. observation (Control group, *n* = 111)	226	EORTC QLQ C30—QLQCR38	1, 3, 6 and 12 mo after surgery	No difference
Sun et al., 2019 [37]	2010–2015	To compare HRQoL after tailored RCT before surgery	5FU + RT/mFOLFOX6+ RT (nCRT group, *n* = 132) vs. mFOLFOX6 alone (nCT group, *n* = 88)	220	EORTC-QLQ C30/CR29	40 mo (median fu after the end of the treatment)	Better global health status, role functioning, and social functioning for the nCT group. Worse stool frequency, flatulence, fecal incontinence, sore skin, and embarrassment for the RT group.
Verweij et al., 2021 [38]	N/A	To compare HRQoL after tailored RCT before surgery	Dose-escalated CRT (5 × 3 Gy boost + CRT) (study group, *n* = 51) vs. CRT alone (control group, *n* = 64)	115	EORTC-QLQ C30/CR29	3, 6, 12, 18, 24 mo after start treatment	No difference
Erlandsson et al., 2021 [39]	1998–2013	To compare HRQoL after tailored RCT before surgery	5 × 5 Gy RT plus surgery within 1 we (SRT, *n* = 51) vs. 5 × 5 Gy RT plus surgery after 4-8 we (SRT-delay, *n* = 57) vs. 25 × 2 Gy RT with surgery after 4-8 we (LRT-delay, *n* = 61)	169	EORTC QLC-C30	4–6 y	No difference
Kosmala et al., 2021 [40]	2006–2010	To compare HRQoL after tailored RCT before surgery	Preoperative CRT followed by TME and postoperative CT (5FU/OX) (*n* = 513) vs. preoperative CRT followed by TME and postoperative CT (5FU) (*n* = 512)	1025	EORTC QLQ C30 EORTC QLQ CR38	After postoperative CT and during fu (1 and 3 y)	No difference
Fokas et al., 2022 [41]	3 y	To compare HRQoL after tailored RCT before surgery	5FU, leucovorin and oxaliplatin before 5FU/OXA CRT (Group A, *n* = 156) vs. CRT before CT (Group B, *n* = 150)	306	EORTC QLQ C30	1 and 3 y	No difference
Ganz et al., 2022 [42]	2004–2010	To compare HRQoL after tailored RCT before surgery	5-FU + RT (*n* = 277) vs. 5-FU + OXA + RT (*n* = 266) vs. CAPE + RT (*n* = 283) vs. CAPE + OXA + RT(n = 286)	1112	SF36, FACT-C	1, 5 y after surgery	No difference
Rouanet et al., 2022 [43]	May 2011–Oct 2014	To compare HRQoL after tailored RCT and induction high-dose chemotherapy	Good responders after induction CT (*n* = 30): surgery (Group A, *n* = 11) or standard RCT plus surgery (Group B, *n* = 19) vs. poor responders after induction CT (*n* = 103): capecitabine (Group C, *n* = 52) or standard RCT (Group D, *n* = 51) before surgery	133	EORTC QLQ-C30	1, 4, 8, 12, 24, 36, 48, and 60 mo	No difference
Dijkstra et al., 2022 [44]	2011–2026	To compare HRQoL after short course RT, CT, TME vs. CRT, TME, and optional adjuvant CT in patients with LARC planned for surgery	Short course RT, CT, TME (study group, *n* = 243) vs. CRT, TME, and optional adjuvant CT (experimental group, *n* = 210)	453	EORTC QLQ C30 EORTC QLQ CR29	36 mo	No difference
Basch et al., 2023 [45]	Jun 2012–Dec 2018	To compare HRQoL after preoperative FOLFOX vs. preoperative 5FUCRT in patients with LARC planned for surgery	Preoperative FOLFOX (*n* = 493) vs. preoperative 5FUCRT (*n* = 447)	940	EuroQoL EQ-5L	12 mo after surgery	Lower rates of fatigue and neuropathy and better sexual function in FOLFOX group
Bascoul-Mollevi et al., 2023 [46]	2012–2017	To compare HRQoL after nCT then CRT, TME, and adjuvant CT vs. CRT, TME, and adjuvant CT in patients with LARC planned for surgery	Neoadjuvant CT (mFOLFIRINOX) then CRT, TME, and adjuvant CT (Study group, *n* = 183) vs. CRT, TME and adjuvant CT (experimental group, *n* = 187)	370	EORTC QLQ C30 EORTC QLQ CR29	1 y, 2 y after treatment	No difference

LAR: Low anterior resection; HRQoL: health-related quality of life; LARS: low anterior resection syndrome; CT: chemotherapy; RT: radiotherapy; RCT: radiochemotherapy; nCT: neoadjuvant CT; TME: total mesorectal excision; N/A: Not Available.

**Table 3 cancers-17-02310-t003:** Articles on patient care strategies included in the review.

Author, Year [Ref.]	Duration of Study	Population	Intervention	Number of Patients	Questionnaire(s)	Times of Assessment	Results
Lee et al., 2013 [47]	Jul 2007–Sep 2011	To compare HRQoL after rehabilitation vs. conventional care in patients who had undergone laparoscopic LAR with defunctioning ileostomy	Rehabilitation (*n* = 52) vs. conventional care (*n* = 46)	98	SF 36	30 days after the operation	No difference
Moug et al., 2019 [48]	Aug 2014–Mar 2016	To compare HRQoL after prehabilitation vs. conventional care in patients planned for nCRT followed by potentially curative surgery	Prehabilitation (*n* = 24) vs. conventional care (*n* = 24)	48	EORTC-QLQ CR29	1-2 w before the operation	No difference
Cuicchi et al., 2020 [49]	Jan 2015–Oct 2015	To compare HRQoL after percutaneous posterior tibial nerve stimulation vs. medical therapy alone in patients who underwent nCRT and LAR for cancer with LARS score ≥ 21 and ileostomy closed at least 18 mo earlier	Percutaneous posterior tibial nerve stimulation (study group, *n* = 6) vs. medical therapy alone (control group, *n* = 6)	12	EORTC-QLQ-C30	1 y	No difference
Yoon et al., 2020 [50]	Jun 2016–Mar 2018	To compare HRQoL after probiotic vs. conventional care in patients undergoing ileostomy closure after LAR for RC	Probiotic (*Lactobacillus plantarum*) (study group, *n* = 17) vs. conventional care (control group, *n* = 19)	36	EORTC-QLQ CR29 EORTC-QLQ-C30	1 w and 3 w following ileostomy reversal	No difference
Morielli et al., 2021 [51]	N/A	To compare HRQoL after exercise vs. conventional care in patients undergoing nCRT	Exercise (study group, *n* = 16) vs. conventional care (control group, *n* = 16)	32	EORTC-QLQ CR29 EORTC-QLQ-C30	Post nCRT, pre-surgery	Worst QoL in the study group (diarrhea *p* = 0.03 and social embarrassment *p* = 0.003)
Su et al., 2021 [52]	Jan 2015–Dec 2015	To compare HRQoL after continuing care bundle vs. conventional care in patients with temporary ileostomy after LAR	Continuing care bundle (study group, *n* = 50) vs. conventional care (control group, *n* = 57)	107	Stoma-QOL	4 and 12 weeks after surgery	Better QoL in the study group (*p* < 0.001)
Asnong et al., 2022 [53]	N/A	To compare HRQoL after PFMT vs. conventional care in patients after LAR and a minimal LARS score of 21/42 at 1 mo after surgery	PFMT (study group, *n* = 50) vs. conventional care (control group, *n* = 54) *PFMT: pelvic floor muscle training*	104	SF-12	1, 4, 6 and 12 mo after the operation	No difference
Van der Heijden et al., 2022 [54]	Oct 2017–Mar 2020	To compare HRQoL after pelvic floor rehabilitation vs. conventional care in patients after LAR	Pelvic floor rehabilitation (study group, *n* = 44) vs. conventional care (control group, *n* = 51)	95	EORTC-QLQ-CR29	3 mo after surgery without temporary stoma or 6 we after stoma closure	No difference (only better HRQoL in patients suffering from fecal incontinence)
Pieniowskui et al., 2023 [55]	May 2016–Nov 2019	To compare HRQoL after TAI vs. conventional care in patients after LAR and a LARS score of 21/42 at 6 mo after surgery	TAI (study group, *n* = 22) vs. conservative treatment (control group, *n* = 23) *TAI: transanal irrigation*	45	EORTC-QLQ-C30	12 mo	Better QoL in the intervention group
Kim et al., 2023 [56]	N/A	To compare HRQoL after bowel function improvement program vs. conventional care	Bowel function improvement program (study group, *n* = 21) vs. conventional care (control group, *n* = 21)	42	EORTC-QLQ-CR29	3 mo after surgery	No difference

LAR: low anterior resection; HRQoL: health-related quality of life; LARS: low anterior resection syndrome; nCRT: neoadjuvant chemoradiotherapy.

### 3.3. Quality of Included Studies and Risk of Bias

The methodological quality of the included RCTs is shown in detail in Appendix A (RoB 2 tool for surgical interventions studies), Appendix A (RoB 2 tool for pre-and/or post-CT and/or RT Studies), and Appendix A (RoB 2 tool for patient-care strategies studies) in the Appendix A. Quality assessment using the RoB 2 tool showed an overall moderate risk of bias in all the studies. Most trials had a low risk of bias in critical areas such as randomization and outcome measurements. However, some concerns were frequently identified, particularly regarding deviations from the intended interventions, incomplete outcome data, and selective reporting. A small number of studies had a high risk of bias in specific areas, including randomization and selective reporting.

### 3.4. Review of Published Studies

#### 3.4.1. Surgical Interventions Studies

The data examination revealed no significant difference in HRQoL between the five (31%) studies that compared the time for stoma closure (early vs. late closure) [23,28,29,30,31], the four (25%) studies that compared the type of colorectal anastomosis (side-to-end vs. straight colorectal vs. colon-J-pouch anastomosis) [18,24,25,26], the three (18%) studies that compared the type of surgical technique (robotic vs. laparoscopic vs. open surgery) [16,21,56], and the study (6%) that analyzed the use of mesh for APR reconstruction [19]. Russell et al. [17] compared the APR and sphincter-saving procedures in patients undergoing TME for rectal cancer. This multicenter study conducted in the UK between 2004 and 2010 included 987 patients with low rectal cancer. Of these, 57% were 59 years or younger, and 66% were male. At the 1-year follow-up, patients who underwent APR reported significantly worse outcomes on the EORTC QLQ-CR38 functional scales, including body image, sexual activities (among males), urinary symptoms, and gastrointestinal symptoms, compared to those who underwent sphincter-saving procedures. Gadan et al. [20] conducted a multicenter study comparing temporary ileostomy and no ileostomy after TME surgery. This study, conducted between 1999 and 2005, included 87 patients diagnosed with rectal cancer. The median age was 62 years (range 32–82 years) in the ileostomy group and 66 years (range 43–84 years) in the non-stoma group. The proportion of males was 61% in the ileostomy group and 52% in the non-stoma group. One year after the primary surgery, the EQ-5D questionnaire revealed no significant difference in overall HRQoL between the two groups. However, those with major LARS reported worse HRQoL outcomes. Bach et al. [27] compared TME surgery with organ preservation via short-course radiotherapy, followed by transanal endoscopic microsurgery. In this multicenter study conducted in the UK between 2012 and 2014, 56 patients with rectal cancer with cT2N0M0 or lower were included. The mean age was 65 years (IQR: 52–79) years in the organ preservation group and 65 years (IQR: 49–83) years in the TME group. The proportion of male patients was 70% in the organ preservation group and 61% in the TME group. The EORTC QLQ-C30 and CR29 demonstrated that patients assigned to the TME group exhibited persistent moderate or large deteriorations in HRQoL, health anxiety, role, and social function at 36 months compared to those in the organ preservation group. The findings of the included studies suggest that the effects of surgical interventions on the quality of life of patients with rectal cancer manifest in both the short and long term. SS procedures have been demonstrated to exert a favorable influence on perceived quality of life in both the short term (3–6 months) and the long term (12 to 36 months) [17,27].

#### 3.4.2. Pre- and/or Post-CT and/or RT Studies

The data examination revealed no significant difference in HRQoL between the seven (46%) studies that compared the type of neoadjuvant therapy (short-course RT vs. long-course CRT, dose-escalated CRT vs. CRT alone, short-course RT with immediate surgery vs. short-course RT with delayed surgery vs. long-course RT with delayed surgery, induction vs. consolidations therapy; induction CT (good responders) + surgery + adjuvant (adj) CT vs. induction CT (good responders) + 50Gy RT/capecitabine + surgery + adj CT vs. induction CT (bad responders) + 50Gy RT/capecitabine + surgery + adj CT vs. induction CT (bad responders) + 60Gy RT/capecitabine + surgery + adj CT, mFOLFIRINOX + CRT + TME + adjCT vs. CRT + TME + adjCT) [32,38,39,41,43,44,46]; between the two (13%) studies that compared the neoadjuvant therapy regimen (capecitabine + RT vs. 5-FU + leucovorin + RT, 5-FU + RT vs. 5-FU + OXA + RT vs. capecitabine + RT vs. capecitabine + oxaliplatin + RT) [34,42]; and between the three (20%) studies that compared the adjuvant regimen (5-FU + leucovorin vs. FOLFOX, capecitabine vs. observational, 5-FU+ oxaliplatin vs. 5-FU alone) [35,36,40]. Wiltink et al. [33] conducted a multicenter study comparing short-course RT followed by TME with TME alone. This study, conducted between 1966 and 1999, included 478 patients, 241 of whom were randomized to the RT arm and 237 to the TME arm. The median age was 62 years (range 43–95) in the RT group and 60 years (range 39–93) in the TME group, with a proportion of males of 62% and 55%, respectively. The study used the EORTC QLQ-C30 and QLQ-CR29 to assess HRQoL and found that bowel function was worse in the RT group. These findings align with those reported by Basch et al. [45], who compared preoperative FOLFOX with preoperative fluorouracil combined with long-course RT. Their study demonstrated that the FOLFOX group experienced lower rates of fatigue and neuropathy as well as better sexual function compared to the 5FU-CRT group. Similarly, Sun et al. [37] conducted a single-center study in China between 2010 and 2015 to compare the HRQoL of patients with rectal cancer undergoing long-course neoadjuvant CRT (nCRT) (5FU + RT/mFOLFOX6 (i.e., modified fluorouracil, leucovorin, and oxaliplatin) + RT) with those undergoing neoadjuvant CT alone (nCT) (mFOLFOX6). This study included 220 patients with rectal cancer. The median age of the patients in the nCRT group was 56 years (range: 27–77) and 55 years (range: 21–77) in the nCT arm. In the nCRT group vs. nCT, 66% and 64% of patients were male. The results of the QLQ-C30 indicated that the nCRT group exhibited poorer global health status, role functioning, and social functioning compared to the nCT group. A substantial body of research has demonstrated the considerable impact of neoadjuvant and adjuvant treatments on the HRQoL of patients diagnosed with rectal cancer. These findings persist both in the immediate postoperative period and over the ensuing long-term course of the disease. In the short term (12 months after surgery), preoperative FOLFOX was associated with an improved quality of life compared to nCRT [45]. However, the deleterious effects of RT were found to be persistent in the long-term setting (40 months and 14 years after surgery) [33,37].

#### 3.4.3. Patient-Care-Strategies Studies

The data examination revealed no significant difference in HRQoL between the four (40%) studies that compared rehabilitation programs (post-surgery rehabilitation vs. pre-surgery rehabilitation) [47,48,53,54], and between the four (40%) studies that compared post-surgery interventions (percutaneous posterior tibial nerve stimulation vs. medical therapy alone, probiotic therapy vs. conventional care, transanal irrigation vs. conventional care, bowel function improvement program vs. conventional care) [49,50,55,56]. Morielli et al. [51] conducted a single-center study in the USA to compare the HRQoL of patients who underwent physical exercise during nCRT with that of those receiving standard care. The study included 32 patients with rectal cancer (RC), with a mean age of 57 years (standard deviation (SD): 12), and 67% were male. The EORTC QLQ-C30 and CR29 questionnaires were used to assess the impact of nCRT physical exercise on patient-reported outcomes. The findings revealed that physical exercise significantly worsened stool frequency, role functioning, emotional functioning, and cognitive functioning compared with usual care. Additionally, following nCRT, exercise was found to exacerbate diarrhea and embarrassment significantly compared to usual care. Su et al. [52] conducted a multicenter study in China involving 107 patients with ileostomy and they compared the effects of a continuing-care bundle with usual care. In the continuing care group, 70% of the patients were male, whereas the usual care group comprised 56% of the patients. The results of the Stoma-QoL assessment indicated that the continuing-care group demonstrated significantly improved self-efficacy, QoL, and satisfaction over time compared with the usual-care group. Pieniowskui et al. [55] conducted a multicenter study examining the impact of transanal irrigation in 45 patients with major LARS. This study ensured comparability between the two groups in terms of age and tumor distance from the anal verge. The findings indicated that TAI was associated with a reduction in LARS symptoms and an improvement in QoL. The assessment of short- and long-term effects in these studies is limited, as the most extended follow-up duration was only 12 months. Overall, patient care strategies, specifically the continuing care bundle and TAI, demonstrated early benefits in quality of life, observable as early as 4 to 12 weeks [52,55].

## 4. Discussion

Over the past few decades, Europe has seen a gradual but consistent increase in the survival rates of patients with rectal cancer, marked by a reduction in mortality from 66% in 2000 to 40% in 2020 [57]. Consequently, optimizing patient HRQoL has become a primary goal in oncology because of the increasing number of patients living with the long-term effects of the disease and its treatment. However, prospective studies of rectal cancer survivors are limited, with many focusing narrowly on individual aspects of functional status such as bowel, urinary, or sexual function. Additionally, there is a scarcity of data from RCTs that often fail to provide effective treatment strategies tailored to the specific needs of patients. To our knowledge, this review represents the first evidence-based synthesis of RCTs examining the influence of surgery, pre/post-CT and/or RT, and patient care on HRQoL in rectal cancer patients in the last decade. These findings suggest the potential for informing shared decision-making frameworks that prioritize the patient’s experience and long-term well-being alongside oncologic outcomes.

### 4.1. Surgical Interventions Studies and HRQoL

Our findings demonstrate that rectal preservation and sphincter-sparing techniques coupled with continuity of care significantly impact HRQoL in these patients. As summarized in Table 1, abdominoperineal resection (as reported by Russell et al.) [17] and rectal resection [27] negatively affect HRQoL compared to sphincter-sparing and organ-preservation surgeries. Commonly affected HRQoL domains include sexual functioning and body image [17], as well as health anxiety and physical, social, emotional, and cognitive functioning [27]. These results align with those of previous meta-analyses [58] and multicenter studies [59], which found that maintaining a stoma-free status and, in particular, preserving the rectum contributes to better HRQoL. Unfortunately, even with intensive total neoadjuvant protocols, the percentage of rectal cancer patients who can spare the rectum after an adequate follow-up period does not exceed 50% of the cases [60] Furthermore, preserving the rectum, especially in cases with high-risk tumor features, can increase the risk of local and distant recurrences [61,62]. When considering SSP, such as low colorectal or coloanal anastomosis, it is important to note that alternatives to abdominoperineal resection are associated with a risk of anastomotic leakage, fecal incontinence, and severe sexual dysfunction. As a result, it is not surprising that a Cochrane review [60] found no clear advantage for restorative surgery in terms of functional outcomes. Moreover, we must consider that the patients enrolled in Russell et al.’s trial [17] were not randomized, with their inclusion being either based on their preoperative sphincter function or their social support network. APR probably remains the best choice, not only when there are no valid oncological alternatives but also in patients with a preoperative sphincter function that has already deteriorated or who are unable to follow an adequate program for the management of low anterior resection syndrome (LARS). The study by Gadan et al. demonstrated that the ostomy itself has been shown to exert a deleterious effect on quality of life over an extended timeframe, specifically, 12 years [20]. Together with the standardization of extraperitoneal rectal cancer surgery, the diverting ileostomy was strongly recommended and became the standard of care after SS procedures in many centers [63]. More recently, a school of thought has emerged that prefers constructing an ileostomy only in patients at high risk of anastomotic fistula [64,65,66]. Ileostomy could reduce the severity of pelvic peritonitis. However, it is itself associated with complications such as dehydration, severe hypokalaemia, and renal failure and requires a second operation for its closure. Moreover, the ileostomy can be related to severe diversion colitis that can contribute to the deterioration of quality of life even months after the stoma closure [67,68]. The results of our review suggest that considering quality of life as an important goal of care, surgeons may plan to avoid ileostomy in cases with a low risk of anastomotic leakage. Consequently, treatment planning should encompass both clinical indicators and patient preferences, thereby facilitating a balanced and individualized decision-making process. The integration of patient-reported outcome data into preoperative consultations has the potential to enhance this approach by empowering patients to select interventions that are aligned with their priorities [69].

### 4.2. Pre- and/or Post-CT and/or RT Studies and HRQoL

In the context of multimodal therapy, comparisons between irradiated (long-course preoperative CRT) and non-irradiated patients (preoperative CT) have been reported in only one RCT [37], which showed that non-irradiated patients exhibited superior HRQoL outcomes, particularly in role and social functioning. Neoadjuvant RT remains a key component of treatment for locally advanced rectal cancer, aiming to reduce the risk of local recurrence and downstage tumor size and, in some cases, achieve a complete clinical and pathological response. Nevertheless, despite the clear oncological benefits of RT, its adverse impact on HRQoL is well-documented [70,71,72]. For example, the Duch trial [33] demonstrated that 14 years after surgery, patients who received short-course pre-operative RT (5 × 5 Gy) had worse cognitive, social, and emotional functioning than those who did not receive RT. These results can be attributed to the well-documented deleterious effect of RT on peripheral nerves. The neurologic damage of RT on pelvic nerves has been investigated in other pelvic cancers (e.g., prostate and endometrial cancer) [73,74]. The nerve injury is probably linked to a radiation-related inflammatory ischemic process with perivascular inflammatory infiltrates or microvasculitis, often inducing chronic neuropathy, resulting in nerve disfunction and consequent deterioration of QoL detectable also many years after treatment completion [75]. This is one of the reasons why Schrag et al. explored neoadjuvant protocols without RT for a subgroup of rectal cancer patients (cT2N+ and cT3 any N, candidates to SSP), demonstrating that preoperative FOLFOX was non-inferior to preoperative CRT with respect to disease-free survival and allows a similar rate of pathological complete response (22.7% vs. 24.7%). Subsequently, Basch et al. reported data on the QoL of patients enrolled in the same trial, showing a better QoL in the group of patients who could avoid RT [45]. RT should also be avoided in the small group of patients with microsatellite unstable rectal cancer: these cases have a formidable opportunity to be cured with exclusive immunotherapy [76,77]. These findings underscore the importance of moving toward adaptive, patient-centered treatment algorithms where the choice of neoadjuvant strategy is guided not only by tumor characteristics but also by patient values, anticipated side effects, and HRQoL priorities. Clinicians should actively discuss the potential short- and long-term trade-offs of different regimens, ideally using decision aids informed by HRQoL data.

### 4.3. Patient-Care-Strategies Studies and HRQoL

Supportive care interventions, particularly continuity-of-care packages, have demonstrated efficacy in improving HRQoL in patients with rectal cancer. Defined as a comprehensive framework designed to ensure coordinated care throughout the treatment journey, continuity-of-care packages, including self-management manuals, telephone follow-ups, and stoma outpatient follow-ups, have proven effective in improving HRQoL after ostomy surgery [52]. Moreover, the benefits of these packages were sustained, even after ostomy closure. These findings underscore the critical importance of optimal stoma care to prevent early complications, many of which can be managed conservatively. Consequently, continuity-of-care packages may help reduce the incidence of stoma-related complications and enhance the outcomes of stoma reversal in patients with temporary stomas. Among patients who have undergone sphincter-sparing surgery for rectal cancer, many have significantly impaired bowel function, known as LARS [78]. Although the treatment of LARS focuses on symptom reduction, no curative treatment currently exists. Transanal irrigation (TAI), in which water is introduced into the intestine to facilitate stool evacuation, has been proposed as a potential treatment option. In our review, TAI was evaluated in one study, which demonstrated its efficacy in improving HRQoL and reducing bowel frequency in patients with rectal cancer and severe LARS [55]. Although the results of this study pertain primarily to patients with severe LARS, they suggest that TAI may also be a viable symptomatic treatment for patients with milder forms of LARS. These results are supported by previous descriptive studies that have shown how TAI after LAR can improve HRQoL and reduce the frequency of bowel movements in patients with rectal cancer [79,80]. These results, related to pre-and post-operative nursing care, seem to be as important as the efforts of technological and technical improvements in the operating room (e.g., laparoscopy and robotic surgery) in improving quality of life for patients with rectal cancer. Care providers should commit to ensuring that decision-makers allocate resources for continuing care strategies. A substantial body of research has demonstrated the effectiveness of exercise interventions during rectal cancer treatment in mitigating treatment-related side effects and enhancing aspects of HRQoL. However, our research indicates that exercise may exacerbate specific symptoms (e.g., diarrhea and embarrassment) and may negatively affect HRQoL during neoadjuvant CRT, making it potentially inappropriate in this clinical setting [51]. The type of exercise prescribed, predominantly aerobic, may be too intensive and inappropriate for this patient cohort. These findings contrast with two recent meta-analyses that reported the positive effects of exercise interventions during active treatment on the functional domains of HRQoL [81,82], emphasizing the need for careful patient selection and customization of care. The incorporation of such strategies into survivorship care plans, in conjunction with patient education regarding available supportive interventions, has the potential to enhance self-efficacy, alleviate symptom burden, and improve post-treatment quality of life. In the contemporary medical landscape, where an increasing number of patients survive critical illnesses but face persistent functional limitations, the implementation of these strategies assumes paramount importance. Emerging digital health tools such as wearable monitors, mobile health applications, and telemedicine platforms offer new opportunities to personalize and optimize follow-up care [83,84,85]. These technologies can facilitate real-time symptom tracking, the early detection of complications, and the dynamic adjustment of care pathways based on patient-reported outcomes. In parallel, personalized follow-up models, guided by risk stratification and patient preference, are being explored as a means to improve efficiency while maintaining quality of care. The integration of such tools into routine clinical practice may represent a promising frontier in enhancing HRQoL in rectal cancer survivors. The results of this systematic review, having the QoL as an essential goal in the treatment of rectal cancer patients, allow us to summarize the following conclusions and suggestions. Rectum-sparing approaches are associated with better QoL. A rectum-sparing approach is possible for up to 50% of patients with a locally advanced microsatellite stable rectal cancer fit for receiving an intensive total neoadjuvant protocol and in the vast majority (if not all) of the patients with microsatellite unstable rectal cancer [60,77]. Care providers should consider a rectum-sparing approach whenever it is oncologically safe and feasible. SSPs without a diverting ileostomy appear to confer a better QoL. Surgeons should stratify tumors and patients in order to individuate cases at low risk for anastomotic leak. In these cases, surgeons might plan a colorectal anastomosis without covering ileostomy. RT is associated with a deterioration of QoL. In cases without the infiltration of other pelvic organs and/or mesorectal fascia and not requiring an APR at first staging, protocols of neoadjuvant chemotherapy might be preferred to the chemoradiation strategy. Continuing care improves the QoL of rectal cancer patients, both with a stoma and with SSP and LARS. Continuing care bundles should be offered to every patient who has experienced rectal resection.

### 4.4. Limitations and Future Directions

Our literature review revealed significant heterogeneity among studies on HRQoL in rectal cancer, which complicates the interpretation of the data accurately. Additionally, many of the included studies have limitations, such as heterogeneity in patient characteristics (e.g., age, gender, and tumor location), and caution is warranted when interpreting their findings. An ideal RCT for analyzing HRQoL in rectal cancer should be controlled, randomized, and, if possible, blinded, with statistical significance determined according to a predefined algorithm [6,86]. Only 19% (8/41) of the studies included in this review met all these criteria. Another critical limitation is the sample size; a small sample may not be representative of the broader population and may lack the statistical power to test hypotheses regarding HRQoL or functional outcomes. Of the 16 surgical RCTs and the 15 pre- and post-CT and/or RT RCTs, 11 (68%) and 14 (93%), respectively, collected data on more than 100 patients. In contrast, only 2 of the 10 patient care RCTs (20%) involved more than 100 patients. This highlights the challenges of conducting representative RCTs on patient care. Additionally, some individual studies included in this review, while methodologically strong, are limited by small cohorts or possible selection bias. For example, the study by Bach et al. [27], which compared TME with organ preservation, provides valuable long-term data but may suffer from limited generalizability due to its small sample size and the greater likelihood that patients with good CRT responses will undergo organ preservation. Such factors must be considered when interpreting individual findings within the broader context of a synthesis. Finally, HRQoL measurements should reflect dimensions that remain consistent throughout a patient’s survival period, emphasizing the importance of considering the time points at which these measurements are taken [76]. A more extended follow-up period yielded more representative HRQoL domains. A total of 6 of the 16 surgical RCTs (37%), 11 of the 15 pre- and post-CT and/or RT RCTs (73%), and only one of the 10 patient care RCTs (20%) collected data for more than 1 year of follow-up. As a result, long-term side effects—such as persistent bowel dysfunction—may be underrepresented. In this context, LARS deserves consideration. LARS affects a significant proportion of patients undergoing sphincter-preserving surgery and is associated with symptoms such as incontinence, urgency, and frequent bowel movements—all of which profoundly impact daily functioning and quality of life [67]. However, general HRQoL questionnaires may not adequately capture the multifaceted burden of LARS, leading to the underestimation of its effects in some studies. The use of disease-specific instruments, such as the LARS score in conjunction with validated HRQoL tools (e.g., EORTC QLQ-C30, CR29), is therefore essential to ensure clinically meaningful assessment. Future research should prioritize incorporating symptom-specific tools and harmonizing the timing of assessment to improve consistency and clinical relevance.

## 5. Conclusions

This systematic review of RCTs highlights the critical role of organ-preserving strategies, including rectal-sparing approaches and continuity-of-care programs, in improving HRQoL in patients with rectal cancer. Although RT adversely affects HRQoL, treatment decisions should be individualized to balance the oncological outcomes with functional preservation. Future prospective RCTs of RC should routinely include HRQoL as a key outcome to ensure a comprehensive evaluation of treatment strategies.

## Figures and Tables

**Figure 1 cancers-17-02310-f001:**
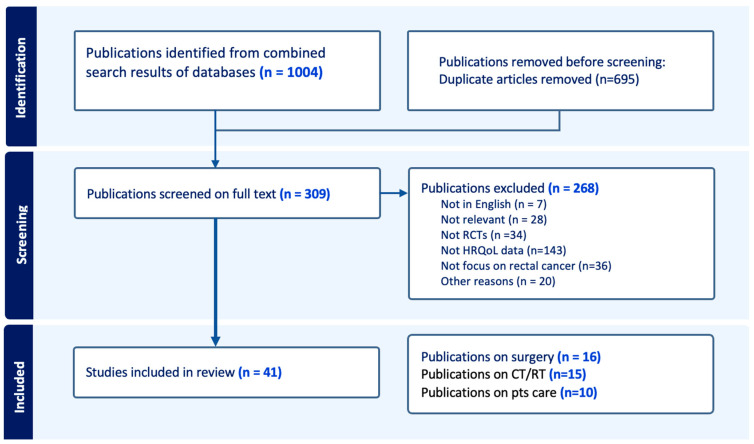
PRISMA flowchart of the included studies.

## Data Availability

No new data were created or analyzed in this study. Data sharing is not applicable to this article.

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
