# Peer review of "Quality of Life in Rectal Cancer Treatments: An Updated Systematic Review of Randomized Controlled Trials (2013–2023)"

_cancers, 2025, doi:10.3390/cancers17142310_

Round 1

Reviewer 1 Report

Comments and Suggestions for Authors

This is an important review with excellent methods of PRO outcomes in rectal cancer. However, the analysis of results needs more work as currently there is very little detail as to the PRO findings, in particular no reference to the differences in terms of timing of treatment/intervention e.g. are these differences acute or late side effects. This needs to be addressed for each intervention for the results to be interesting clinically.

The detail of tumour size and distance from the verge is not relevant as for any RCT these would be randomised if relevant and can be removed or moved to table if these are felt to be important findings with reference to the PRO data. 

Limitations - radiation should have been used as a search term for US English. 

Author Response

Reviewer 1:

This is an important review with excellent methods of PRO outcomes in rectal cancer. However, the analysis of results needs more work as currently there is very little detail as to the PRO findings, in particular no reference to the differences in terms of timing of treatment/intervention e.g. are these differences acute or late side effects. This needs to be addressed for each intervention for the results to be interesting clinically.

We thank the reviewer for this important observation. We would like to clarify that the timing of patient-reported outcomes (PROs) is addressed throughout the Results section (3.3.1–3.3.3), where we specify the follow-up durations (e.g., 1 year, 36 months) and, when applicable, whether the effects are persistent or occur post-treatment. Additionally, the timing of PRO assessments is reported in Tables 1–3, helping to distinguish between acute and late effects. We have slightly revised the result section (“The reviewed studies reported patient-reported outcome (PRO) data collected at various follow-up stages, ranging from the early postoperative period (e.g., four weeks) to long-term assessments (e.g., 12 years). This enabled the identification of acute, treatment-related symptoms as well as persistent, long-term impairments in HRQoL.” ) and results section (surgery; therapy; and Patient Care Strategies Studies). Regarding the discussion section we have already included the timing in the limitations “Finally, HRQoL measurements should reflect dimensions that remain consistent throughout a patient's survival period, emphasizing the importance of considering the time points at which these measurements are taken(73). A longer follow-up period yielded more representative HRQoL domains. Six of the 16 surgical RCTs (37%), 11 of the 15 pre- and post-CT and/or RT RCTs (73%), and only one of the 10 patient care RCTs (20%) collected data for more than 1 year of follow-up.”

The detail of tumour size and distance from the verge is not relevant as for any RCT these would be randomised if relevant and can be removed or moved to table if these are felt to be important findings with reference to the PRO data. 

We have adopted the suggestion and removed the details on “distance from the anal verge” from the body of the text. These details are now exclusively included in the summary tables (Table 1-3) to facilitate comparative reference where deemed relevant. This approach maintains the methodological rigor of RCTs while respecting the principles of randomization and avoiding the burden of excessive detail.

Limitations - radiation should have been used as a search term for US English. 

We thank the reviewer for their thoughtful comment regarding the inclusion of "radiation" as a search term. Our search strategy was designed to capture studies specifically investigating the therapeutic use of radiotherapy in rectal cancer rather than general references to radiation exposure. To ensure a focused and clinically relevant selection of randomized controlled trials evaluating patient-reported outcomes, we used the MeSH term "Radiotherapy," combined with treatment-related terms such as "therapy" and "treatment."

While adding "radiation" as a free-text term (e.g., title or abstract) might retrieve additional articles, we opted for a more specific strategy to maximize the relevance of the results and minimize irrelevant articles from non-therapeutic contexts. We are confident that our approach is methodologically sound and aligned with the objectives of our review.

For transparency, we have clarified this rationale in the revised Methods section (“The term "Radiotherapy" was used as a MeSH term in our research to specifically capture therapeutic interventions, while the broader term "radiation" was excluded to maintain focus and specificity.”  “Data collection and analysis”).

Reviewer 2 Report

Comments and Suggestions for Authors

I have reviewed the manuscript and found it to be well written, methodologically sound, and free from major issues. However, to enhance the clarity and readability of the work, there are several minor concerns that warrant the authors’ careful attention.

Minor concerns:

  • While the paper correctly avoids meta-analysis due to heterogeneity, this could be better supported by a quantitative heterogeneity index (e.g., I²). Consider briefly discussing the implications of this heterogeneity on the strength of conclusions.
  • The conclusions summarize key findings well, but critical appraisal of conflicting evidence, especially in RT vs CT effects, could be expanded.
  • More emphasis on patient-centered decision-making frameworks or clinical translation would improve its practical relevance.
  • While the study uses RoB 2, the narrative summary is vague (“moderate risk” generally stated). It would help to provide a summary table or chart showing which domains (e.g., outcome reporting, randomization) had the highest concerns.
  • Some tables (e.g., Table 1–3) are detailed but references to these in the main text are infrequent or abrupt.
  • Mention of Supplementary Figures (1a, 2a, 3a) in Section 3.2 should also include brief descriptions of what they show.
Comments on the Quality of English Language

While overall well-written, there are occasional grammatical slips (e.g., “compared than surgery alone” should be “compared to surgery alone”). A final round of English language proofreading is recommended.

Author Response

Reviewer 2:

I have reviewed the manuscript and found it to be well written, methodologically sound, and free from major issues. However, to enhance the clarity and readability of the work, there are several minor concerns that warrant the authors’ careful attention.

Minor concerns:

  • While the paper correctly avoids meta-analysis due to heterogeneity, this could be better supported by a quantitative heterogeneity index (e.g., I²). Consider briefly discussing the implications of this heterogeneity on the strength of conclusions.

Response : Thank you for your comment. We agree that quantifying heterogeneity can be useful in many systematic reviews. However, in our study, the considerable clinical and methodological heterogeneity among the included studies-in terms of interventions (surgery, chemoradiation therapy, and supportive care), outcome measures (different HRQoL instruments), patient populations, and duration of follow-up-precluded the use of a meaningful I² index.

Given the high level of variability, any statistical measure of heterogeneity would not have provided reliable insights and could have been misleading. Instead, we chose to address heterogeneity qualitatively, through a structured narrative summary and a stratified presentation of results according to intervention type and time of evaluation. We have now clarified this point in the discussion to strengthen the rationale for our methodological approach ( “Due to this significant variability, the calculation of a quantitative heterogeneity index (e.g., I²) was deemed inappropriate, as it would not provide a reliable measure of inconsistency across studies with such a wide range of methodological approaches. Therefore, a structured narrative synthesis was adopted to ensure clarity and preserve the clinical relevance of the findings.”

  • The conclusions summarize key findings well, but critical appraisal of conflicting evidence, especially in RT vs CT effects, could be expanded.
  • Thank you for the suggestion. We revised the conclusion section to include a more thorough critical appraisal of the conflicting evidence regarding the effects of radiotherapy (RT) versus chemotherapy (CT) on quality of life (HRQoL). In particular, we highlighted that although RT represents a standard in neoadjuvant treatment of locally advanced rectal cancer for locoregional control of disease, several randomized trials have documented a persistent negative impact of RT on cognitive, sexual, and social function even in the long term (e.g., studies by Wiltink et al. and Sun et al.). In contrast, chemotherapy regimens without RT (e.g., preoperative FOLFOX) have been shown to be associated with fewer side effects and better HRQoL in the short term, while maintaining comparable oncologic efficacy in selected patients (e.g., study by Basch et al.). We also discussed how the choice of treatment should consider a balance between oncologic efficacy and preservation of quality of life, taking into account patient preferences and individual risk of recurrence. These aspects were incorporated into the discussion section to reinforce the importance of a personalized approach.

  • More emphasis on patient-centered decision-making frameworks or clinical translation would improve its practical relevance.
  • Thank you for your valuable suggestion. We fully agree that enhancing the practical relevance of our findings is essential. In response, we have revised the Discussion section to place greater emphasis on patient-centered decision-making frameworks and clinical translation. Specifically, we now highlight how HRQoL data can inform shared decision-making in the context of surgical planning and neoadjuvant/adjuvant treatment choices. We also underscore the importance of integrating patient preferences and values when selecting organ-preserving strategies or supportive care interventions. These additions aim to better connect our findings with real-world clinical practice and the evolving focus on personalized, value-based oncology.
  • While the study uses RoB 2, the narrative summary is vague (“moderate risk” generally stated). It would help to provide a summary table or chart showing which domains (e.g., outcome reporting, randomization) had the highest concerns.

We thank the reviewer for this helpful comment. We apologize for the omission — the detailed RoB 2 domain-level assessments were not included in the initial submission. We have now added a comprehensive summary table to the Supplementary Materials, specifying the level of risk for each domain (e.g., randomization process, deviations from intended interventions, missing outcome data, measurement of the outcome, selection of the reported result). We hope this addition provides greater clarity and transparency regarding the methodological quality of the included studies.

  • Some tables (e.g., Table 1–3) are detailed but references to these in the main text are infrequent or abrupt.

We have revised the main text to ensure clearer and more frequent references to Tables 1–3. These references have been integrated more smoothly into the narrative to guide the reader through the findings of each intervention category (surgical, neoadjuvant/adjuvant, and patient care strategies).

  • Mention of Supplementary Figures (1a, 2a, 3a) in Section 3.2 should also include brief descriptions of what they show.

In Section 3.2, we have added brief descriptions of Supplementary Figures 1a, 2a, and 3a to clarify what each figure illustrates. These figures visually summarize the timing and domains of HRQoL effects across the three categories of intervention, supporting the narrative presentation of short- and long-term impacts.

While overall well-written, there are occasional grammatical slips (e.g., “compared than surgery alone” should be “compared to surgery alone”). A final round of English language proofreading is recommended.

We extend our gratitude to the reviewer for this insightful observation. The manuscript has been meticulously revised to rectify all grammatical inaccuracies, including the previously noted instance of incorrect usage ("compared than surgery alone," now corrected to "compared to surgery alone"). A final round of meticulous English language proofreading has been completed to ensure clarity and correctness throughout the text.

Reviewer 3 Report

Comments and Suggestions for Authors

The Web of Science database was not searched. When the keyword “rectal neoplasms” was used alone, 1379 articles were found.

How was the bias in publications eliminated? I recommend checking the existence of publication bias with a funnel plot.
The article was presented as a systematic review. I recommend strengthening the findings by adding a meta-analysis.

The publications are presented in a single table, making evaluation and reading difficult. I recommend that the table be simplified or submitted as an additional file.

Author Response

Reviewer 3:

The Web of Science database was not searched. When the keyword “rectal neoplasms” was used alone, 1379 articles were found.

We appreciate the reviewer’s comment regarding the inclusion of the Web of Science database. In our study, we opted to use Scopus as one of our primary databases, as it provides broader coverage across scientific disciplines compared to Web of Science. To ensure the completeness of our search strategy, we cross-checked the yield from Scopus, which identified 352 additional records. Among these, five studies were not initially included in our dataset and were subsequently screened. However, after full-text evaluation, all five were excluded based on the predefined eligibility criteria due to lack of relevance to our review objectives. Therefore, we are confident that the use of Scopus has effectively compensated for the omission of Web of Science and has not impacted the comprehensiveness of our review. This methodological choice has now been clarified in the revised manuscript.

How was the bias in publications eliminated? I recommend checking the existence of publication bias with a funnel plot.
The article was presented as a systematic review. I recommend strengthening the findings by adding a meta-analysis.

We appreciate the reviewer’s suggestion regarding the use of a funnel plot. However, due to the substantial clinical and methodological heterogeneity among the included studies—encompassing different interventions (surgical, medical, and supportive care), outcome measures (various HRQoL instruments), follow-up durations, and patient populations—a meta-analysis was not performed (Sterne JAC, Egger M, Moher D, Boutron I (editors). Chapter 10: Addressing reporting biases. In: Higgins JPT, Churchill R, Chandler J, Cumpston MS (editors), Cochrane Handbook for Systematic Reviews of Interventions version 5.2.0 (updated June 2017), Cochrane, 2017. Available from www.training.cochrane.org/handbook). As a result, effect sizes are not directly comparable across studies, and generating a funnel plot would not yield interpretable or meaningful results in this context. We have clarified this in the Methods section

The publications are presented in a single table, making evaluation and reading difficult. I recommend that the table be simplified or submitted as an additional file.

Thank you for your comment. In response to your suggestion, we have simplified the main table by removing data on sex, age, distance from the anal verge, and study design. These details have been included in the supplementary material to facilitate a clearer and more concise presentation in the main manuscript.

Reviewer 4 Report

Comments and Suggestions for Authors

Negro S. et al.: Quality of Life in Rectal Cancer Treatments: An Updated Sys- 2 tematic Review of Randomized Controlled Trials (2013-2023)

This systematic review to systematically analyze the impact of rectal cancer treatment on HRQoL. A total of 41 studies comprised 9,240 patients were recruited. This systematic review of RCTs underscores the importance of organ-sparing strate- 30 gies, such as rectum-sparing approaches and continuity-of-care packages, in improving HRQoL in 31 patients with rectal cancer.

I have a few remarks:

General comment:

I would like to express my gratitude to the authors of this paper for their efforts; it is both interesting and elegantly written.

Methods:

  • Can the authors explain why they chose the time span 2013 to 2023?
  • Were only studies with multimodal therapy taken or all of them?
  • How do the authors deal with the high heterogeneity of the included studies, which excluded a meta-analysis? Is the qualitative summary sufficiently robust as a result?
  • Why was an institutional ethics approval waived despite the systematic nature of the study, and how are ethical issues of the included original studies dealt with?
  • How meaningful is the use of different QoL questionnaires (Quality of Life) for the comparison of the study results, and was the validity of these measurement instruments taken into account?

Results:

  1. How meaningful are the results if several studies show no significant differences in HRQoL between different surgical methods? Could this be due to an insufficient sample size or methodological differences?
  2. To what extent do the results of Russell et al. (17) regarding poorer HRQoL after APR reflect actual differences between surgical techniques, and how were potential confounding factors controlled for?
  3. What role does low anterior resection syndrome (LARS) play in the assessment of quality of life after ileostomy, and how might this influence the interpretation of the study results by Gadan et al. (20)?
  4. How robust are the findings of Bach et al. (27) on the comparison of TME and organ preservation, particularly with regard to the long-term consequences for quality of life, and what limitations could restrict their validity?
  5. To what extent could differences in patient characteristics (e.g. age, gender, tumour location) influence the HRQoL results in the different studies, and how was this taken into account in the analysis?

Discussion:

The literature is discussed very nicely in the Discussions section.

The following points will be discussed:

  1. Provide clearer recommendations for future research, such as standardizing HRQoL measurement timepoints, using consistent validated instruments, and incorporating patient preferences. Also consider adding a forward-looking perspective on emerging therapies or digital health tools that might improve HRQoL.
  2. Instead of simply listing studies, discuss how findings complement or contradict each other. For example, if two meta-analyses show differing impacts of radiotherapy on HRQoL, explore possible reasons for the discrepancy.

Author Response

Reviewer 4:

This systematic review to systematically analyze the impact of rectal cancer treatment on HRQoL. A total of 41 studies comprised 9,240 patients were recruited. This systematic review of RCTs underscores the importance of organ-sparing strate- 30 gies, such as rectum-sparing approaches and continuity-of-care packages, in improving HRQoL in 31 patients with rectal cancer.

I have a few remarks:

General comment:

I would like to express my gratitude to the authors of this paper for their efforts; it is both interesting and elegantly written.

Methods:

  • Can the authors explain why they chose the time span 2013 to 2023?
  • Were only studies with multimodal therapy taken or all of them?
  • How do the authors deal with the high heterogeneity of the included studies, which excluded a meta-analysis? Is the qualitative summary sufficiently robust as a result?
  • Why was an institutional ethics approval waived despite the systematic nature of the study, and how are ethical issues of the included original studies dealt with?
  • How meaningful is the use of different QoL questionnaires (Quality of Life) for the comparison of the study results, and was the validity of these measurement instruments taken into account?

Results:

  1. How meaningful are the results if several studies show no significant differences in HRQoL between different surgical methods? Could this be due to an insufficient sample size or methodological differences?
  2. To what extent do the results of Russell et al. (17) regarding poorer HRQoL after APR reflect actual differences between surgical techniques, and how were potential confounding factors controlled for?
  3. What role does low anterior resection syndrome (LARS) play in the assessment of quality of life after ileostomy, and how might this influence the interpretation of the study results by Gadan et al. (20)?
  4. How robust are the findings of Bach et al. (27) on the comparison of TME and organ preservation, particularly with regard to the long-term consequences for quality of life, and what limitations could restrict their validity?
  5. To what extent could differences in patient characteristics (e.g. age, gender, tumour location) influence the HRQoL results in the different studies, and how was this taken into account in the analysis?

Discussion:

The literature is discussed very nicely in the Discussions section.

The following points will be discussed:

  1. Provide clearer recommendations for future research, such as standardizing HRQoL measurement timepoints, using consistent validated instruments, and incorporating patient preferences. Also consider adding a forward-looking perspective on emerging therapies or digital health tools that might improve HRQoL.
  2. Instead of simply listing studies, discuss how findings complement or contradict each other. For example, if two meta-analyses show differing impacts of radiotherapy on HRQoL, explore possible reasons for the discrepancy.

Author Response:
We sincerely thank the reviewer for the thoughtful and constructive feedback, as well as for the positive evaluation of our work. We address each point below:

Methods:

Can the authors explain why they chose the time span 2013 to 2023?

We selected the 2013–2023 time frame to focus on randomized controlled trials (RCTs) that reflect contemporary treatment practices, HRQoL measurement tools, and supportive care strategies. Treatments, surgical techniques, and patient-centered outcomes have evolved significantly in the past decade, and including older studies might have introduced outdated or less relevant practices. We explain this in the methods section

 Were only studies with multimodal therapy taken or all of them?
All RCTs evaluating HRQoL outcomes in rectal cancer patients were considered, regardless of whether they employed multimodal therapy. Studies were grouped into three predefined categories: surgical interventions, pre- and/or post-CT and/or RT, and patient care strategies. This allowed us to capture a comprehensive picture of treatment-related HRQoL across different modalities.

  • How do the authors deal with the high heterogeneity of the included studies, which excluded a meta-analysis? Is the qualitative summary sufficiently robust as a result?
    We acknowledge the substantial heterogeneity in study designs, interventions, outcome tools, and follow-up durations, which precluded a quantitative meta-analysis. Instead, we conducted a structured qualitative synthesis based on predefined variables and outcomes. We also explicitly addressed this issue in the Methods and Limitations sections, and discussed the implications for generalizability and clinical translation.
  • Why was an institutional ethics approval waived despite the systematic nature of the study, and how are ethical issues of the included original studies dealt with?
    As this was a systematic review of previously published studies, no new data were collected and no human participants were directly involved. Therefore, ethics committee approval was not required. All included studies were peer-reviewed RCTs, and we relied on the ethical approvals obtained by the original authors, as described in their respective publications.
  • How meaningful is the use of different QoL questionnaires for the comparison of the study results, and was the validity of these measurement instruments taken into account?

We acknowledge the use of different validated instruments (e.g., EORTC QLQ-C30, CR29, SF-36, EQ-5D) across studies. While this limits direct comparability, all included tools are validated for oncologic or colorectal populations and widely used in clinical research. We have noted this heterogeneity in the Discussion and Limitations sections and emphasized the need for standardization in future studies.

Results:

  1. How meaningful are the results if several studies show no significant differences in HRQoL between different surgical methods? Could this be due to an insufficient sample size or methodological differences?
    Yes, the lack of statistical significance in some studies may be attributable to insufficient sample sizes, short follow-up durations, or methodological variability. We highlighted this in our discussion of individual studies and in the Limitations section, where we emphasized the need for larger, adequately powered RCTs.
  2. To what extent do the results of Russell et al. (17) regarding poorer HRQoL after APR reflect actual differences between surgical techniques, and how were potential confounding factors controlled for?

The RCT by Russell et al. attempted to control for confounding through randomization; however, residual confounders such as baseline functional status or patient preference may still influence outcomes. We interpreted their findings cautiously and contextualized them with those of other RCTs and cohort studies in the Discussion.

  1. What role does low anterior resection syndrome (LARS) play in the assessment of quality of life after ileostomy, and how might this influence the interpretation of the study results by Gadan et al. (20)?
    LARS plays a significant role in HRQoL and is often underappreciated. The Gadan study demonstrated that even years after stoma reversal, patients may experience poor HRQoL due to LARS. We have expanded on this point in the discussion and limitation section to emphasize the long-term implications of functional bowel disorders.
  2. How robust are the findings of Bach et al. (27) on the comparison of TME and organ preservation, particularly with regard to the long-term consequences for quality of life, and what limitations could restrict their validity?
    The findings of Bach et al. offer valuable insights, particularly due to the randomized design and long follow-up. However, the relatively small sample size and potential selection bias (patients with good response to CRT were more likely to undergo organ preservation) may limit generalizability. We discussed this nuance in our synthesis
  3. 5. To what extent could differences in patient characteristics (e.g. age, gender, tumour location) influence the HRQoL results in the different studies, and how was this taken into account in the analysis?
    We recognize that differences in patient characteristics could significantly affect HRQoL outcomes. Most included RCTs stratified results by relevant variables or used multivariate analysis. However, given the heterogeneity and narrative nature of our synthesis, we were not able to fully control for these confounders across studies. This is acknowledged as a limitation

Discussion:

  1. Provide clearer recommendations for future research, such as standardizing HRQoL measurement timepoints, using consistent validated instruments, and incorporating patient preferences. Also consider adding a forward-looking perspective on emerging therapies or digital health tools that might improve HRQoL.
    We thank the reviewer for this insightful suggestion. We have revised the final paragraphs of the Discussion to include specific recommendations for standardizing timepoints, using validated tools, and integrating patient preferences into research and care. Additionally, we introduced a brief discussion on digital tools and emerging therapies, such as wearable monitors and personalized follow-up models.
  2. Instead of simply listing studies, discuss how findings complement or contradict each other. For example, if two meta-analyses show differing impacts of radiotherapy on HRQoL, explore possible reasons for the discrepancy.
    We appreciate this recommendation and have revised the discussion of surgical and RT-related studies to highlight areas of concordance and divergence. Where discrepancies were identified, we proposed possible explanations based on study design, patient selection, or outcome measures.

Round 2

Reviewer 3 Report

Comments and Suggestions for Authors

Responses to criticisms by the author are acceptable.